# The Unfolded Protein Response Is a Major Driver of LCN2 Expression in BCR–ABL- and JAK2V617F-Positive MPN

**DOI:** 10.3390/cancers13164210

**Published:** 2021-08-21

**Authors:** Stefan Tillmann, Kathrin Olschok, Sarah K. Schröder, Marlena Bütow, Julian Baumeister, Milena Kalmer, Vera Preußger, Barbora Weinbergerova, Kim Kricheldorf, Jiri Mayer, Blanka Kubesova, Zdenek Racil, Martina Wessiepe, Jörg Eschweiler, Susanne Isfort, Tim H. Brümmendorf, Walter Becker, Mirle Schemionek, Ralf Weiskirchen, Steffen Koschmieder, Nicolas Chatain

**Affiliations:** 1Department of Hematology, Oncology, Hemostaseology and Stem Cell Transplantation, Faculty of Medicine, RWTH Aachen University, 520674 Aachen, Germany; sttillmann@ukaachen.de (S.T.); kolschok@ukaachen.de (K.O.); mbuetow@ukaachen.de (M.B.); jbaumeister@ukaachen.de (J.B.); mikalmer@ukaachen.de (M.K.); kkricheldorf@ukaachen.de (K.K.); sisfort@ukaachen.de (S.I.); tbruemmendorf@ukaachen.de (T.H.B.); mschemionek@ukaachen.de (M.S.); skoschmieder@ukaachen.de (S.K.); 2Center for Integrated Oncology Aachen Bonn Cologne Düsseldorf (CIO ABCD), 52074 Aachen, Germany; saschroeder@ukaachen.de (S.K.S.); rweiskirchen@ukaachen.de (R.W.); 3Institute of Molecular Pathobiochemistry, Experimental Gene Therapy and Clinical Chemistry (IFMPEGKC), Faculty of Medicine, RWTH Aachen University, 52074 Aachen, Germany; 4Institute of Pharmacology and Toxicology, Faculty of Medicine, RWTH Aachen University, 52074 Aachen, Germany; vera.preussger@freenet.de (V.P.); wbecker@ukaachen.de (W.B.); 5Department of Internal Medicine, Hematology and Oncology, Masaryk University and University Hospital Brno, 625 00 Brno, Czech Republic; Weinbergerova.Barbora@fnbrno.cz (B.W.); Mayer.Jiri@fnbrno.cz (J.M.); Kubesova.Blanka@fnbrno.cz (B.K.); Zdenek.Racil@uhkt.cz (Z.R.); 6Institute of Hematology and Blood Transfusion, 12820 Prague, Czech Republic; 7Institute of Transfusion Medicine, Faculty of Medicine, RWTH Aachen University, 52074 Aachen, Germany; mwessiepe@ukaachen.de; 8Department of Orthopedic Surgery, Faculty of Medicine, RWTH Aachen University, 52074 Aachen, Germany; joeschweiler@ukaachen.de

**Keywords:** Lipocalin 2 (LCN2), NGAL, ER stress, UPR, MPN, BCR-ABL, JAK2V617F, IRE1

## Abstract

**Simple Summary:**

Lipocalin 2 (LCN2) is a proinflammatory mediator increased in the blood of patients with myeloproliferative neoplasms (MPN) and other hematologic malignancies, significantly contributing to MPN disease initiation and progression. Here, we investigated the underlying mechanisms of LCN2 overexpression in MPN. We found a strong correlation between BCR–ABL and JAK2V617F driver oncogenes and LCN2 expression. Furthermore, LCN2 expression is strongly induced by endoplasmic reticulum (ER) stress, independent of oncogenic kinase activity. We identified the IRE1–JNK–NF-κB–C/EBP axis as a major mediator of ER stress-induced LCN2 expression. Our findings provide novel insights into the regulation of LCN2 and present a basis for innovative, targeted treatment approaches in MPN.

**Abstract:**

Lipocalin 2 (LCN2), a proinflammatory mediator, is involved in the pathogenesis of myeloproliferative neoplasms (MPN). Here, we investigated the molecular mechanisms of LCN2 overexpression in MPN. LCN2 mRNA expression was 20-fold upregulated in peripheral blood (PB) mononuclear cells of chronic myeloid leukemia (CML) and myelofibrosis (MF) patients vs. healthy controls. In addition, LCN2 serum levels were significantly increased in polycythemia vera (PV) and MF and positively correlated with JAK2V617F and mutated CALR allele burden and neutrophil counts. Mechanistically, we identified endoplasmic reticulum (ER) stress and the unfolded protein response (UPR) as a main driver of LCN2 expression in BCR-ABL- and JAK2V617F-positive 32D cells. The UPR inducer thapsigargin increased LCN2 expression >100-fold, and this was not affected by kinase inhibition of BCR-ABL or JAK2V617F. Interestingly, inhibition of the UPR regulators inositol-requiring enzyme 1 (IRE1) and c-Jun *N*-terminal kinase (JNK) significantly reduced thapsigargin-induced LCN2 RNA and protein expression, and luciferase promoter assays identified nuclear factor kappa B (NF-κB) and CCAAT binding protein (C/EBP) as critical regulators of mLCN2 transcription. In conclusion, the IRE1–JNK-NF-κB–C/EBP axis is a major driver of LCN2 expression in MPN, and targeting UPR and LCN2 may represent a promising novel therapeutic approach in MPN.

## 1. Introduction

Myeloproliferative Neoplasms (MPN) are clonal chronic malignancies of the bone marrow (BM), characterized by rapid expansion of blood cells and accompanied by an inflammatory bone marrow microenvironment [1]. They are closely associated with the MPN driver oncogenes, such as BCR-ABL in chronic myeloid leukemia (CML) [2], JAK2V617F in polycythemia vera (PV), essential thrombocythemia (ET) and myelofibrosis (MF) [3,4], and calreticulin (CALR) frameshift mutations in ET and MF [5,6]. Inflammatory cytokines such as interferons, interleukin-1β (IL-1β) or IL-8 fuel disease progression [7] and may contribute to the initial transformation of the malignant clone [8,9]. While the development of tyrosine kinase inhibitors (TKI) (e.g., imatinib and ruxolitinib) allows specific targeting of these oncogenes and reduce symptom burden and inflammation [7], a better understanding of the interplay between the malignant clone and its microenvironment including circulating cytokines is needed.

Lipocalin 2 (LCN2) or neutrophil granulocytes associated lipocalin (NGAL) is one of the inflammatory factors increased in MPN patients [10]. It is a 25 kDa secreted glycoprotein, originally found in neutrophils as part of the innate immune response [11]. In the hematopoietic system, LCN2 is predominantly found in myeloid cells [10]. In MPN, LCN2 was shown to be increased in the serum of CML patients [12] and to be required for the development of CML in mice [13]. Furthermore, LCN2 favors fibrotic transformation of the BM microenvironment [14], and is secreted by JAK2V617F-positive cells, leading to increased reactive oxygen species (ROS) levels and subsequent DNA damage in healthy hematopoietic cells [15]. Besides its role in the BM, an increase of LCN2 expression has been observed in response to stress signals. Tumor necrosis factor alpha (TNF-α) and IL-1β increased *LCN2* expression in the liver [16], and lipopolysaccharides (LPS) induced LCN2 expression in the lung and liver [17]. Moreover, *LCN2* expression was linked to endoplasmic reticulum (ER) stress in the kidney, caused by proteinuria [18].

One of the central cellular mechanisms in response to ER stress and accumulation of misfolded proteins is the unfolded protein response (UPR). High metabolic or secretory activity, oxidative stress, nutrient deprivation, hypoxia, and inflammation are typical triggers of the UPR [19,20]. Due to its importance in different hematopoietic neoplasms, UPR has evolved into a potential therapeutic target in recent years [21]. In myeloid malignancies, the inositol-requiring enzyme 1 (IRE1) and protein kinase RNA-like endoplasmic reticulum kinase (PERK) pathways are of prominent interest [21]. BCR-ABL-positive cells have been shown to rely on PERK signaling to cope with high metabolic activity and to develop imatinib resistance [22]. In addition, IRE1 activity is increased in patients with acute myeloid leukemia (AML), and its inhibition leads to apoptosis of malignant cells [23].

Despite the known role of LCN2 in pathogenesis, the molecular mechanisms leading to increased *LCN2* expression in hematopoietic malignancies are still poorly understood. In this study, we investigated the pathways that are essential for LCN2 expression in BCR-ABL- and JAK2V617F-positive MPN.

## 2. Materials and Methods

### 2.1. Primary Patient Samples

Peripheral blood and bone marrow samples were obtained from MPN patients at the Department of Hematology, Oncology, Hemostaseology and Stem Cell Transplantation of RWTH Aachen University, or from fully anonymized healthy individuals at the Department of Transfusion Medicine at RWTH Aachen University, both after written informed consent, as approved by the local ethics committee (EK127/12, EK206/09 and EK099/14, respectively) (Appendix A). Additional samples were obtained from the University Hospital Brno, Czech Republic, within the CZECH MPN project MIND after written informed consent, as approved by the multicenter and local ethics committee of the University Hospital Brno on 10 April 2013 (Appendix A). Femoral heads served as a source of healthy bone marrow mononuclear cells (BMMCs) and were obtained from the Orthopedics department from the RWTH Aachen University, each after written informed consent and in compliance with the local ethics committee (EK300/13). Human PBMCs and BMMCs were isolated by density gradient centrifugation using Pancoll reagent (PAN-Biotech, Aidenbach, Germany).

### 2.2. Cell Culture

32D and HEK293-T cells were obtained from the German collection of microorganisms and cell culture (DSMZ, Braunschweig, Germany) and cultured in RPMI-1640 or DMEM medium (Pan biotech, Aidenbach, Germany), respectively. Both media were supplemented with 10% FCS (Pan biotech, Aidenbach, Germany) and 1% Penicillin/Streptomycin (Gibco, Grand Island, NY, USA). Additionally, WEHI3B supernatant (10%) was added to RPMI-1640 medium as an IL-3 source for the 32D cells.

The pMY-IRES-GFP vector was used as the empty vector control or containing cDNA for BCR-ABL or JAK2V617F. Production of viral particles and transduction of 32D cells were performed as previously described [24].

### 2.3. Cytokines and Chemical Compounds

For all experiments, 32D cells were washed with phosphate-buffered saline (PBS) and seeded at a density of 1 × 10^6^ cells/mL in 12-well plates. Murine TNF-α (ImmunoTools, Friesoythe, Germany), human IL-1β (ImmunoTools, Friesoythe, Germany), murine IFN-α (Miltenyi Biotec, Bergisch Gladbach, Germany), rat IFN-γ (Peprotech, Hamburg, Germany), thapsigargin (Tha, Sigma-Aldrich, St. Louis, MO, USA) and brefeldin A (BFA, Sigma-Aldrich, St. Louis, USA) were used for stimulation. For inhibition of signaling pathways, imatinib and ruxolitinib were purchased from LC Laboratories (Woburn, MA, USA). GSK2606414, STF-083010, Kira-6, LY-2409881, SB203580, JNK-IN-8 and T5224 were purchased from Hycultec (Beutelsbach, Germany) and applied as described in the figure legends. Afterwards, cells were washed with PBS and harvested for RNA isolation or protein extraction.

### 2.4. Multiplex Immunoassay for Serum Samples

ProcartaPlex™ Multiplex Immunoassay (ThermoFisher Scientific, Waltham, MA, USA) was performed according to the manufacturer’s protocol. Serum samples of MPN patients were collected by centrifugation of blood samples for 10 min at 1000× *g*. Serum samples were subsequently stored at −80 °C. Before use, the samples were centrifuged for 10 min at 10,000× *g* and diluted 1:100 in Universal Assay Buffer provided by the supplier.

### 2.5. RT-qPCR

RNA isolation, cDNA synthesis, and RT-qPCR were conducted as described before [24]. Gene expression was calculated as a percentage of Glycerinaldehyd-3-phosphate-dehydrogenase (GAPDH) expression. All primers used in this study are listed in Appendix A.

### 2.6. SDS Page and Western Blotting

Western blot analysis was performed as previously described [24]. Detection was performed using the Fusion SL (PeqLab, Erlangen, Germany). Antibodies used for this study are listed in Appendix A. Densitometry analysis was performed using ImageJ software 1.53i. Densitometry of all Western blots are available in Appendix A.

### 2.7. Propidium Iodide Staining

1 × 10^6^ cells were treated with varying concentrations of Tha and stained for dead cells using propidium iodide (PI; Sigma-Aldrich, St. Louis, MO, USA) in a 1:1000 dilution. Samples were subsequently analyzed using a FACS Gallios (Beckmann Coulter, Krefeld, Germany).

### 2.8. Generation of Promoter Constructs

Human LCN2 promoter fragments were PCR amplified from genomic DNA from HepG2 cells JumpStart RED Accutaq DNA Polymerase (Sigma Aldrich, Taufkirchen, Germany) and ligated into the pGL3 vector (Promega, Madison, WI, USA) via engineered restriction sites (KpnI, NcoI). For murine constructs, fragments of the *Lcn2* promoter from genomic DNA of C57BL/6 mice were amplified by PCR using DREAMtaq polymerase (Thermo Fisher Scientific, Waltham, MA, USA). Primers used for amplification contained restriction sites for *KpnI* and *HindIII* flanking the promoter fragments. The PCR products were cloned into a TOPO cloning vector (Invitrogen, Thermo Fisher Scientific, Waltham, MA, USA) according to the manufacturer’s protocol. The TOPO vectors containing the promoter fragments were subsequently digested using *KpnI* and *HindIII* (both from New England Biolabs, Ipswich, MA, USA), and the fragments were integrated into the pGL3 vector using T4 ligase (New England Biolabs, Ipswich, MA, USA). The primers used for amplification are listed in Appendix A.

### 2.9. Site Directed Mutagenesis

To change potential transcription factor binding sites (TFBS) in our promoter constructs, the Q5-Site Directed Mutagenesis Kit (New England Biolabs, Ipswich, MA, USA) was used according to the manufacturer´s protocol. Primers used for amplification are listed in Appendix A.

### 2.10. Luciferase Assay

For luciferase promoter assays, 32D were electroporated with the described pGL3 vectors containing one of the promoter fragments in combination with the pRL-null vector using the Neon transfection system (1500 V, 10 ms, 3 pulses; Thermo Fisher Scientific). After 18 h, cells were stimulated with Tha for 6 h and consecutively analyzed for firefly and *Renilla* luciferase expression using the Dual Glo Stop and Glo assay Kit (Promega, Madison, WI, USA), according to the manufacturer´s protocol. For measuring luciferase activity, a Victor X3 plate reader (PerkinElmer, Waltham, MA, USA) was used.

HEK293-T cells were seeded in a 6-well plate, grown to 80% confluence, and transfected with the aforementioned vectors using TransIT transfection reagent (Mirus, Madison, WI, USA). After 24 h, cells were transferred to 96-well plates and allowed to reattach overnight. The following day, cells were stimulated with Tha for 6 h and luciferase activity was assessed as described for 32D cells.

### 2.11. Statistical Analysis

Statistical analysis was performed using GraphPad Prism software 5.0 (GraphPad Software, La Jolla, CA, USA). For comparison of two groups under normal distribution, two-tailed Student’s *t*-test was conducted. For analysis of more than two groups, one-way ANOVA with Dunnett’s post hoc-test was applied. When data did not follow a Gaussian distribution, comparison of two groups was done by the Mann–Whitney test; for comparison of more groups, the Kruskal–Wallis test following Dunns post hoc-test was conducted. To analyze correlations, a linear regression model was applied. Data are shown as mean with standard deviation (SD). Significant differences are shown by asterisks, indicating *p*-values of: * = *p* < 0.05; ** = *p* < 0.01; *** = *p* < 0.001. All experiments were performed at least three times, unless indicated differently.

## 3. Results

### 3.1. Differences in LCN2 Expression Are Linked Both to Oncogenes and Cellular Phenotype

As *LCN2* expression has previously been described to be upregulated in patients with MPN [25], we first wanted to confirm and extend these data. Therefore, the expression of *LCN2* mRNA was assessed in peripheral blood mononuclear cells (PBMCs, Figure 1A) or bone marrow mononuclear cells (BMMCs, Figure 1B) isolated from MPN patients and healthy donors (HD). While a significant increase of *LCN2* mRNA levels in PBMCs of patients with CML, PV, and MF compared to HD was found, no differences were observed in ET patients. In BMMCs, we did not observe any difference between HD, CML, ET, and MF, while PV had significantly lower *LCN2* expression.

As LCN2 is highly expressed in granulocytes [10], the cell composition of the analyzed BMMC samples was examined. Hence, we performed RT-qPCR analysis of myeloperoxidase (*MPO*), erythropoietin receptor (*EPOR*), and myeloproliferative leukemia protein (*MPL*) expression to correlate *LCN2* expression with the quantity of cells of the granulomonocytic, erythrocytic, and megakaryocytic lineage, respectively (Figure 1C). Neither *MPO* nor *EPOR* showed a significant difference in expression to HD, while *MPL* expression was increased in PV, ET, and MF patients. As expected, *MPO* expression was highest in CML samples, *MPL* expression was highest in ET and MF, and *EPOR* expression was highest in PV patients. While no correlation between *LCN2* and *MPO* was found, *MPL* and *EPOR* showed a significant negative correlation to *LCN2* mRNA levels (Figure 1C–E).

Next, we performed a multiplex immunoassay to measure LCN2 protein in the blood of PV, ET, and MF patients or HD (Figure 2A). LCN2 was significantly increased in the PB of patients with PV as well as MF. When sorted by oncogene (Figure 2B), JAK2V617F but not CALR mutations were associated with higher LCN2 serum protein. We also separated patients by current treatment at the time of sampling (Appendix A), showing significantly lower LCN2 protein when patients were treated with hydroxyurea, TKI, or IFN-α, as compared to no treatment (= watchful waiting). Especially, untreated PV and MF patients showed increased LCN2 in comparison to HD (Appendix A). Notably, we detected no influence of gender or age on LCN2 (Appendix A), but a strong correlation between allelic burden and the concentration of leukocytes or neutrophils (Appendix A).

To further investigate the cause of increased LCN2 protein, we correlated LCN2 with allele burden of JAK2V617F or CALR (Figure 2C). In both cases, we found a significant positive correlation between allele burden and LCN2 serum level. In addition, there was a positive correlation between neutrophils in the PB and LCN2 protein levels in serum (Figure 2D). Additionally, a similar correlation was observed for leukocytes in general, as well as platelets (Appendix A).

### 3.2. ER Stress Induces Lcn2 mRNA Expression Independent of Oncogene Activity

As a myeloid cell model, murine 32D cells were stably transduced with BCR–ABL, JAK2V617F or the respective empty vector (EV) and subsequently analyzed for *Lcn2* expression (Figure 3A). Both oncogenes led to a significant increase in *Lcn2* mRNA, which was highest in BCR–ABL-transduced cells. To detect Lcn2 protein, cells were treated for 8 h with brefeldin A (BFA) to block secretion and accumulate Lcn2 in the cell. Only BCR–ABL-transduced cells showed an increase in Lcn2 protein (Figure 3B).

To identify further factors that affect *Lcn2* expression, we stimulated the cells with the inflammatory cytokines TNF-α, IL-1β, IFN-α, and IFN-γ as well as thapsigargin (Tha), an inducer of ER stress and the associated UPR (Figure 3C). While none of the cytokines caused a significant change in *Lcn2* expression, Tha markedly increased *Lcn2* mRNA levels in all cells independent of the oncogene. Induction of Lcn2 expression was comparable when using BFA and lower after tunicamycin, a classical inducer of ER stress (Appendix A). To further investigate the effect of ER stress, we subjected the cells to increasing concentrations of Tha for 24 h and analyzed *Lcn2* mRNA and protein expression (Figure 3D,E). In BCR-ABL and JAK2V617F cells, 30 nM Tha caused a strong increase in *Lcn2* mRNA expression (Figure 3D). Lcn2 protein increased from 30 nM up to 1000 nM in a concentration-dependent manner (Figure 3E; Appendix A). In comparison, 32D EV cells only showed a mild increase in *Lcn2* mRNA expression at 30 nM Tha, and protein was detected at a concentration of 100 nM Tha. It should be noted that neither of these concentrations led to an increase in the dead cell fraction (PI-positive)** in the analyzed cells after 24 h (Appendix A). After 48 h, higher concentrations of Tha increased the amount of dead cells in all cell lines. Next, we analyzed IRE1α as a marker for UPR activation and found increased protein levels in the EV cells after 30 nM Tha treatment and already at 10 nM in the BCR-ABL- and JAK2V617F-positive cells (Figure 3E). Densitometry analysis underscored these findings, with EV cells only showing significant differences at 300 nM, while BCR–ABL cells already increased significantly at 10 nM and JAK2V617F at 30 nM (Appendix A). The cell number was significantly lower after 48 h in the presence of 10 nM of Tha, indicating UPR-induced cell death and/or cell cycle arrest (Appendix A) [26]. Hence, in the following experiments, 30 nM Tha and exposure times of less than 24 h were applied.

To elucidate if kinase activities of BCR–ABL and JAK2V617F influences ER stress-mediated induction of *Lcn2*, we conducted co-treatments of the 32D BCR-ABL and JAK2V617F cells with Tha in combination with increasing concentrations of imatinib or ruxolitinib, respectively (Figure 3F,G). While TKIs alone had no effect on *Lcn2* mRNA (Figure 3F) or protein (Figure 3G), rising concentrations of TKIs in combination with Tha treatment caused a decrease in Lcn2 protein. Counterintuitively, this result was not reflected on mRNA level, as increased *Lcn2* by Tha treatment was not reduced by addition of TKI (Figure 3F). Decreased phosphorylation of CRKL (for BCR–ABL) or STAT5 (for JAK2V617F) confirmed the inhibitory effect of the TKIs (Figure 3G). Additionally, the presence of Tha alone caused a reduction of CRKL and STAT5 phospho- and total protein level. Again, IRE1α protein increased upon Tha treatment, indicating UPR activation, while TKI treatment did not (Figure 3G).

### 3.3. UPR-Induced Lcn2 Expression Is Regulated by the IRE1α—JNK Axis

To further investigate the pathways underlying the induction of Lcn2 by ER stress, we combined Tha and the PERK inhibitor GSK2606414 (GSK), the IRE1α nuclease activity inhibitor STF-083010 (STF), or the IRE1α nuclease and kinase activity inhibitor Kira-6, respectively (Figure 4A). While GSK and STF did not alter *Lcn2* mRNA levels, Kira-6 reduced them in all cell lines.

Since inhibition of IRE1 kinase but not nuclease activity altered *Lcn2* expression, we analyzed downstream targets of IRE1, using the IKK inhibitor LY-2409881 (LY), p38 inhibitor SB203580 (SB), or the c-Jun *N*-terminal kinase (JNK) inhibitor JNK-IN-8 (Figure 4B). Inhibition of IKK did not change *Lcn2* mRNA expression. However, expression was significantly reduced by JNK inhibition. Conversely, p38 inhibition increased *Lcn2* levels (Figure 4B). To verify our previous findings at the protein level, we conducted Western blot analysis with Kira–6 or JNK–IN-8 treated cells in combination with Tha (Figure 4C,D). As expected, Tha strongly induced UPR, demonstrated by increased IRE1α protein and subsequent activation of JNK. JNK kinase activity was demonstrated by phosphorylation of its downstream target c-Jun (Figure 4D). LCN2 protein was induced upon Tha application, more pronounced in the oncogene expressing 32D cells, and was reduced after addition of Kira-6 or JNK-IN-8. Co-treatment with Kira-6 or JNK-IN-8 reduced JNK and c-Jun phosphorylation, respectively (Figure 4C,D). In addition, combined treatment of Kira-6 and Tha decreased IRE1α protein in 32D EV and BCR–ABL but not JAK2V617F cells.

To translate our findings to MPN patient samples, we incubated PBMCs of three PV patients, carrying the JAK2V617F mutation, for 24 h with Kira-6 or JNK-IN-8 and analyzed changes in *LCN2* mRNA expression (Figure 4E). In all three samples, inhibition of IRE1 or JNK significantly reduced *LCN2* expression.

### 3.4. LCN2 Promoter Activity upon ER Stress Is Dependent on C/EBP and NF-κB Promoter Binding Sites

To gain deeper insights into the activity of the *LCN2* promoter during ER stress, we generated fragments of the human and murine *LCN2* promoter and integrated each into a luciferase reporter vector (Figure 5A). The murine constructs were transiently transfected by electroporation into the different 32D cell lines and stimulated with Tha (Figure 5B–D). We found a significant increase in promoter activity in all fragments down to 189 bp. No induction was observed using the 132 bp promoter fragment after Tha treatment of the 32D EV and BCR-ABL, and only a very mild induction in 32D JAK2V617F cells (Figure 5B–D). In general, 32D JAK2V617F showed the highest baseline response rate of the three cell lines. It should be noted that the baseline activity for the 132 bp construct was particularly low compared to the larger fragments in the BCR–ABL and JAK2V617F.

Next, we transfected human HEK293-T cells with reporter constructs carrying promoter fragments similar in size to the murine constructs (Figure 5E). In line with our findings in 32D cells, we observed significantly increased promoter activity after Tha stimulation in all fragments down to 189 bp. In contrast, the 189 bp fragment showed a strongly reduced response to Tha stimulation, and the 132 bp promoter fragment was inactive. In summary, reporter gene inducibility by Tha was mainly driven by a promotor element located between 412-189 bp.

The analysis of the sequence between 412-189 bp revealed potential TFBS for activator protein 1 (AP-1), nuclear factor kappa B (NF-κB), and CCAAT enhancer binding protein (C/EBP). We mutated these binding sites to random sequences in the human 412 bp construct, which were no longer predicted to bind the respective transcription factors (Appendix A). These mutated constructs were transfected into HEK293-T cells and promoter activity was measured after Tha treatment (Figure 5F). Mutation of the AP-1 binding sites did not cause any changes in Tha-induced activation. This was reconfirmed by chemical inhibition of AP-1 using T5224, resulting in stable inhibition of IL-1β but not LCN2 (Appendix A Appendix A). However, mutation of the NF-κB binding site resulted in a much lower induction of promoter activity after Tha treatment. Most prominently, the loss of the second C/EBP binding site resulted in complete loss of promoter activity. These findings indicate an important role of NF-κB and C/EBP binding sites for ER stress dependent *LCN2* induction.

## 4. Discussion

LCN2 was shown to play a critical role in disease development and progression of MPN [13,14,15]. To counteract its elevated expression in chronic myeloid neoplasms, a better understanding of the regulatory mechanisms responsible for LCN2 induction is needed. In this study, we demonstrated a strong connection of UPR activation and *LCN2* expression in BCR-ABL- and JAK2V617F-positive MPN, independent of oncogenic kinase activity. Furthermore, induction of *LCN2* expression is mediated by the IRE1α–JNK–NF-κB–C/EBP axis and can be altered by interfering with this pathway.

Under physiological conditions, *LCN2* expression is high in the BM, but low in PB. This difference can be explained by changes in *LCN2* expression during granulocyte maturation. Granulocytic progenitors, mainly residing in the BM, show high *LCN2* expression with accumulated LCN2 protein in specific granules [27]. Consequently, mature granulocytes were shown to contain high amounts of LCN2 protein, but low levels of *LCN2* mRNA. This is also reflected in our data, as we found a marked difference between PB and BM expression of *LCN2* in HD samples. Therefore, differences between HD and MPN patient samples may be due to changes in the cellular composition, manipulated by the oncogene. This is also supported by the fact that *LCN2* expression was different between PV, ET, and MF, despite all of them harboring the JAK2V617F mutation. The observation that *LCN2* negatively correlated with *MPL* and *EPOR* is also in line with these findings. On the contrary, *MPO* did not correlate with *LCN2* expression, which might be explained by the fact that the analyzed cell fraction (mononuclear cells separated by density gradient centrifugation) excludes terminally differentiated granulocytic cells. Some very recent studies on different hematological malignancies reported that LCN2 protein strongly correlates with neutrophil count in PB and BM [28] and differs with diagnosis [29]. Of note, the markers used in this study are not sufficiently specific to determine exact cell populations expressing *LCN2*, as they may also differ from patient to patient. However, the correlations point towards cell fractions not expressing EPOR or MPL as main contributors to elevated LCN2 levels in MPN.

Our analysis of serum from MPN patients confirmed these findings, but also leukocyte and platelet counts in general correlated with LCN2 serum protein. An increase in these cell types is associated with inflammatory conditions such as MPN. In this regard, a contribution of the BM microenvironment is also possible and should be investigated more closely in the future. In addition, LCN2 serum protein correlated with both JAK2V617F as well as CALR allele burden. However, LCN2 was only significantly increased in JAK2-mutated patients compared to healthy individuals. JAK2 mutations in ET and MF are associated with more complications and lower survival rates compared to CALR [30], which may be reflected by increased LCN2 and other proteins. Furthermore, neutrophil and leukocyte counts strongly correlated with allele burden in general. Therefore, increased LCN2 serum protein might be in part a consequence of leukocytosis in these patients. In accordance with these findings, any kind of cytoreductive therapy was associated with significantly lower LCN2 serum levels.

Aside from cellular composition and driving oncogenes, we demonstrated a strong connection between the induction of *Lcn2* expression and ER stress in our 32D cell model. While BCR–ABL and JAK2V617F led to an increase in *Lcn2* mRNA, induction by Tha exceeded this increase in the analyzed cells. In addition to the increased expression, accelerated secretion of LCN2 is imaginable, explaining increased serum levels in MPN patients. Overall, it can be assumed that the other MPN-related driver mutations (e.g., mutated CALR or MPL) might also affect LCN2 expression directly or indirectly. The interplay between the driving oncogene and ER stress needs closer investigation in the future.

Interestingly, BCR–ABL cells appeared to be especially sensitive to increasing dosages of Tha. This might be due to a generally high amount of metabolic activity caused by this oncogene, making them more sensitive to additional ER stress. Furthermore, inhibition of kinase activity by imatinib or ruxolitinib did not affect *LCN2* mRNA expression after Tha treatment. However, increasing concentrations of TKI decreased LCN2 protein in the presence of Tha. Since we also observed decreased CRKL and STAT5 protein, it is likely that loss of oncogenic activity in addition to UPR induction caused a decrease of survival signaling in these cells, reducing protein synthesis and possibly triggering proteasomal degradation of Lcn2, while the UPR-driven transcriptional machinery is still active. It should be noted that Tha treatment alone caused a decrease in STAT5 protein, probably resulting from general reduction in protein synthesis caused by the UPR [26]. Nevertheless, Tha did not induce cell death at any concentration within 24 h, indicating that cells were under stress but viable under these conditions.

It is worth noting that we detected no significant increase of *Lcn2* upon inflammatory cytokine stimulation, such as TNF-α or IL-1β, which are described to increase expression in liver or endothelial cells [16,31]. These findings suggest that *LCN2* expression in hematopoietic cells is mainly driven by intrinsic stress and less dependent on paracrine induction. However, we only applied four different cytokines, known to regulate *LCN2* expression, and a plethora of additional inflammatory factors are highly expressed in MPN [32]. As BCR–ABL and mutant JAK2 cause an increase in metabolic activity and protein synthesis, these cells are dependent on UPR activity to cope with the resulting ER stress [33].

The connection between ER stress and *LCN2* expression has already been described in other malignancies like prostate cancer and chronic kidney disease [18,34]. However, little is known about its role in MPN. In AML, the IRE1–XBP1 axis was described to be highly active and important for cell survival [35]. In addition, c-Jun was shown to be overexpressed in AML cells, directly inducing the expression of UPR target genes (e.g., XBP1 and ATF-4), thereby allowing the cells to resolve ER stress [36]. In line with these data, we observed that inhibition of JNK, phosphorylating and activating c-Jun, reduced *Lcn2* expression. In BCR–ABL-positive ALL patients, high XBP Protein levels and activation of IRE1 are associated with poor outcome [37]. Recent studies revealed that loss of LCN2 causes much stronger UPR activation upon ER stress, indicating that LCN2 expression might be involved in resolving or managing ER stress [38]. Taken together, increased *LCN2* expression is not only a result of oncogenic activity but also dependent on ER stress levels and microenvironmental conditions as well as the predominant cell types involved in the disease.

Upon induction of ER stress, all branches of the UPR become strongly activated and an intricate network of downstream pathways is set in motion. Despite its complexity, our study identified IRE1 as the main sensor molecule driving *LCN2* expression upon ER stress. Furthermore, JNK plays a key role in signal transduction and is a well-known target of IRE1 [39]. While a link between JNK activity and *LCN2* expression has previously been reported, it was associated to inflammatory cytokines, including TNF-α and IL-1β [40]. Both, the TNF-α receptor (TNFR) and IRE1 share a common adaptor protein in the TNF receptor-associated factor 2 (TRAF2), explaining the activation of JNK by both pathways. We observed that inhibition of IRE1 kinase activity caused a much stronger decrease in *LCN2* expression than inhibition of JNK, suggesting that other downstream pathways are involved.

Accordingly, our analysis of TFBS in the promoter region of both murine and human *LCN2* genes supports this hypothesis. The NF-κB and C/EBP binding sites in close proximity to the translational start site (at −250 and −160 bp, respectively) were shown to be essential for ER stress-induced gene expression. NF-κB has previously been shown to be involved in *LCN2* regulation in adipocytes [41]. In contrast, inhibition of IKK2, an upstream kinase of NF-κB, by LY-2409881 did not alter *LCN2* expression in our 32D cell system. Therefore, NF-κB may be induced by a different pathway that involves JNK. Oxidative stress was reported to induce JNK activity, leading to the accumulation of beta-transducin repeats-containing protein (b-TrCP), part of the SCF-b-TrCP E3 ligase complex [42,43]. This complex mediates IκB ubiquitination and degradation and activation of NF-κB. As Tha is an inducer of oxidative stress [44], NF-κB may still be activated in our current model via JNK despite IKK2 inhibition. Of note, as Tha affects calcium homeostasis in both the ER and the cytosol, it is possible that calcium-dependent mechanisms may be involved in the regulation of LCN2 expression as well.

On a similar note, CEBPα and CEBPβ have been shown to be regulated by JNK activity [45]. Hence, both binding sites appear to be involved in the induction of the *LCN2* gene, and it is likely that multiple transcription factors are involved in *LCN2* regulation. To our surprise, AP-1 was not involved in gene regulation in both systems. Neither mutation of the AP-1 binding sites nor its inhibition by T5224 altered promoter activity or *LCN2* mRNA expression, respectively. This is in contrast to studies that reported the necessity of AP-1 activity for *LCN2* expression in kidney fibroblasts following LPS treatment [46]. However, our experiments with the human promoter constructs were conducted in HEK293-T cells and therefore might not be fully transferable to hematopoietic cells. Especially, the lack of an oncogene may affect the basal expression levels in this system. Nevertheless, our results point towards important regulatory regions in the *LCN2* promoter and provide a basis for further investigations on transcription factor-mediated (e.g., C/EBPα/β) and UPR-induced *LCN2* gene activity.

Noteworthy, inhibition of p38 signaling increased *LCN2* mRNA levels in addition to Tha. JNK and p38 belong to the family of mitogen-activated protein kinases (MAPK), and their activity is strongly interconnected. Therefore, inhibition of p38 might cause a hyper-activation of JNK as a compensatory effect, resulting in higher LCN2 expression.

Altogether, our findings suggest the IRE1–JNK–NF-κB–C/EBP axis as the main driver of ER stress induced *LCN2* expression in our MPN model. The LCN2 protein may have a protective function upon excessive ER stress in hematopoietic cells, as has been previously suggested for hepatocytes [38]. Additionally, high *LCN2* expression is not directly induced by oncogenic activity, but rather a secondary effect of the cellular transformation and cell composition mediated by the malignant cells. Further investigations are needed to clarify the role of LCN2 in UPR management of malignant cells in MPN.

## 5. Conclusions

Taken together, increased *LCN2* expression in MPN is an indirect consequence of oncogenic signaling in the malignant clones. The amount of LCN2 found in the blood is greatly affected by the composition of the hematopoietic cell compartment. In myeloid BM cells, ER stress strongly drives *LCN2* expression independent of a driving oncogene. With the UPR as the dedicated mechanism to deal with ER stress, the IRE1–JNK axis is the main cause of this increase in *LCN2* expression, with both C/EBPs and NF-κB being involved in transcriptional regulation.

## Figures and Tables

**Figure 1 cancers-13-04210-f001:**
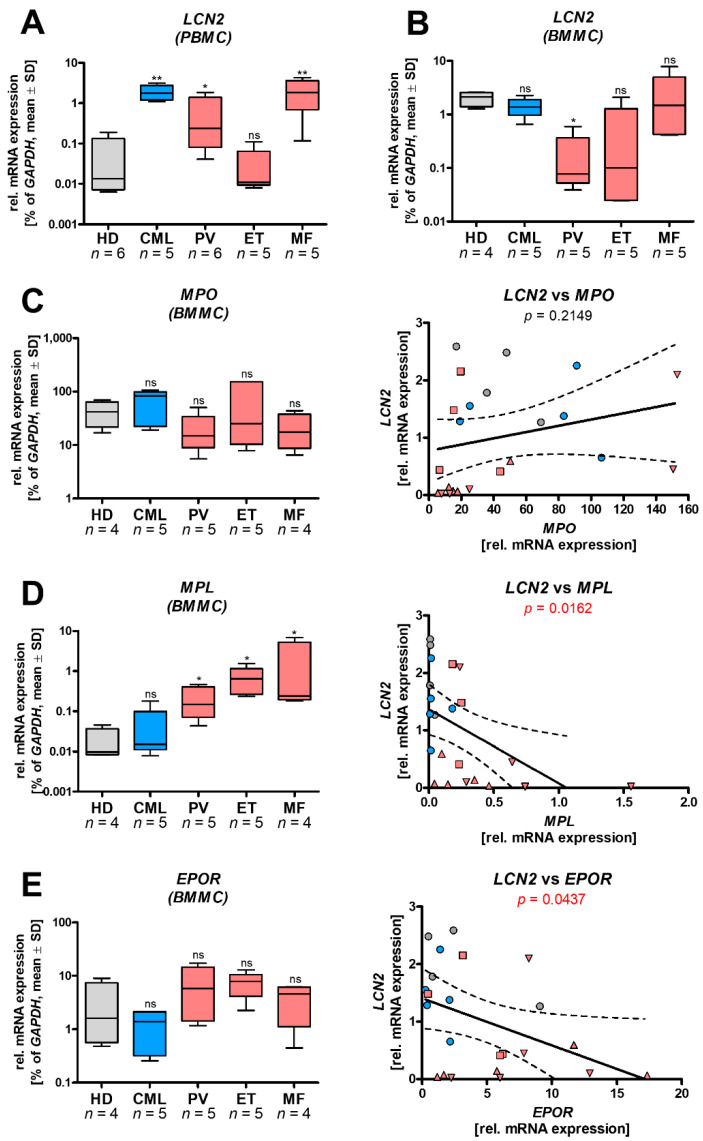
Differences in LCN2 expression between MPN entities. RT-qPCR analysis of (**A**) peripheral blood mononuclear cells (PBMC) or (**B**) bone marrow mononuclear cells (BMMC) from healthy donors (HD) or patients diagnosed with chronic myeloid leukemia (CML), polycythemia vera (PV), essential thrombocythemia (ET) or myelofibrosis (MF). Colors indicating the driver oncogene: blue: BCR-ABL, red: JAK2V617F. Mann–Whitney U test, comparing to HD. (**C**–**E**) left: RT-qPCR analysis of bone marrow mononuclear cells (BMMCs) of patients with the indicated diagnosis. Mann–Whitney U test, comparing to HD. right: Correlation between indicated mRNA data of all samples measured in (**B**–**E**). grey dots = HD, blue dots = CML, red downwards triangle = PV, red upwards triangle = ET, red square = MF. Linear regression analysis. Asterisks indicate *p*-values of: * = *p* < 0.05; ** = *p* < 0.01. ns—not significant.

**Figure 2 cancers-13-04210-f002:**
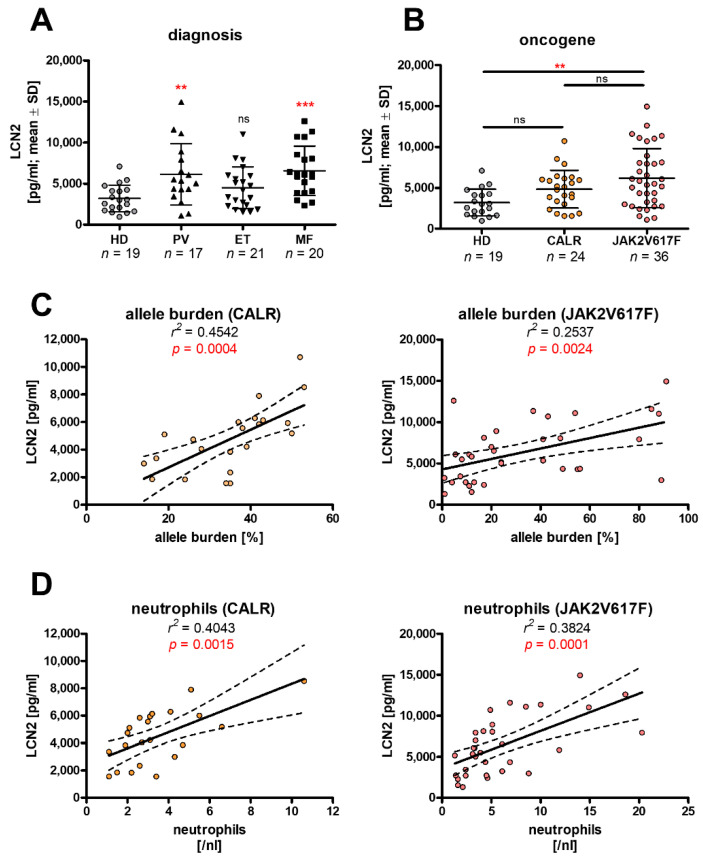
LCN2 serum levels are increased in PV and MF patients. LCN2 serum levels of healthy donors (HD) or patients diagnosed with polycythemia vera (PV), essential thrombocythemia (ET) or myelofibrosis (MF) were assessed from PB using the ProcartaPlex™ Multiplex Immunoassay and grouped by (**A**) diagnosis (Mann–Whitney U test) or (**B**) oncogene. (Kruskal–Wallis followed by Dunn´s test) (**C**,**D**) Correlation between LCN2 serum levels and allele burden (**C**) or neutrophil count (**D**) for patients carrying a CALR mutation (left) or JAK2 mutation (right). Linear regression analysis. Asterisks indicate *p*-values of: ** = *p* < 0.01; *** = *p* < 0.001. ns—not significant.

**Figure 3 cancers-13-04210-f003:**
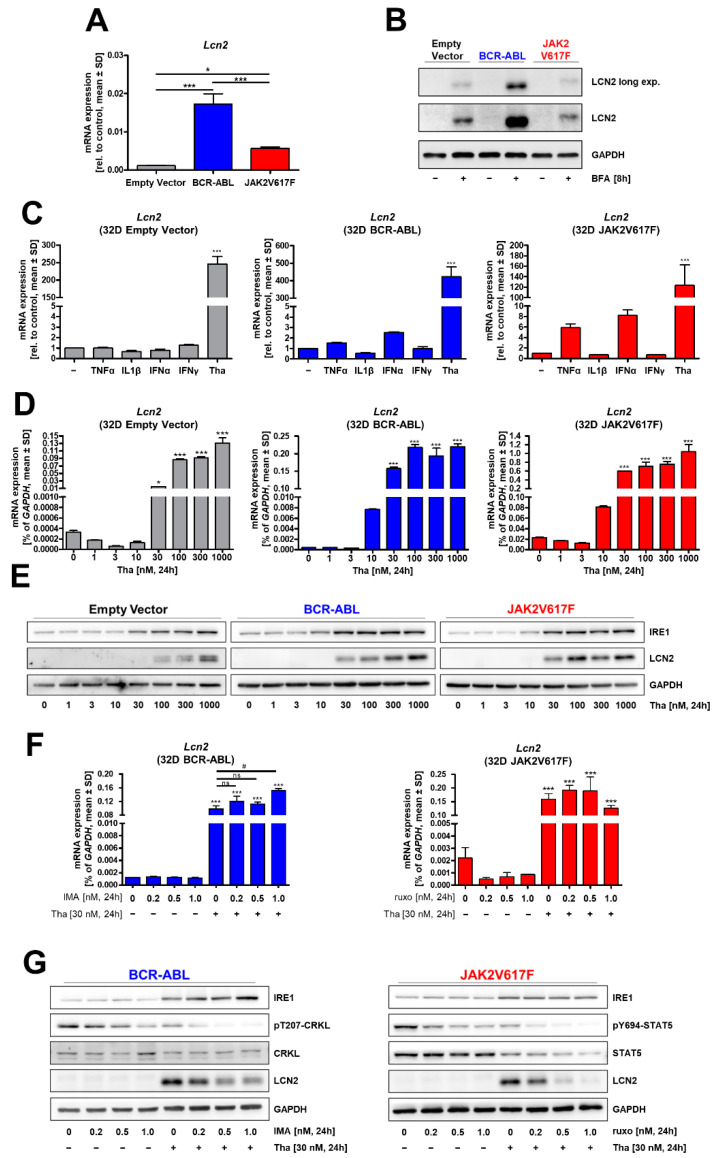
LCN2 expression is induced by ER stress. (**A**) RT–qPCR analysis of mRNA isolated from the stably transduced 32D cells expressing BCR-ABL (blue), JAK2V617F (red) or the empty vector (grey). *n* = 3, ANOVA followed by multiple comparison tests. (**B**) Western blot analysis of 32D cell lysates after 8 h treatment with 2 µg/mL brefeldin A (BFA) to show accumulation of Lcn2 protein. *n* = 3 (**C**) RT-qPCR analysis of 32D cells treated with 20 ng/mL TNF-α, 50 ng/mL IL-1β, 200 U IFN-α, 200 U IFN-γ or 100 nM thapsigargin (Tha) for 24 h. *n* = 3, ANOVA followed by Dunnett’s test. (**D**,**E**) RT–qPCR (**D**) and Western blot analysis (**E**) of indicated 32D cells treated for 24 h with increasing concentrations of Tha. *n* = 3, ANOVA followed by Dunnett’s test. (**F**,**G**) RT-qPCR (**F**) and Western blot analysis (**G**) of 32D BCR-ABL or JAK2V617F cells treated with 30 nM Tha combined with increasing concentrations of imatinib (IMA) or ruxolitinib (ruxo), respectively. *n* = 3. ANOVA followed by Dunnett’s test. Asterisks indicate *p*-values of: * = *p* < 0.05; *** = *p* < 0.001.

**Figure 4 cancers-13-04210-f004:**
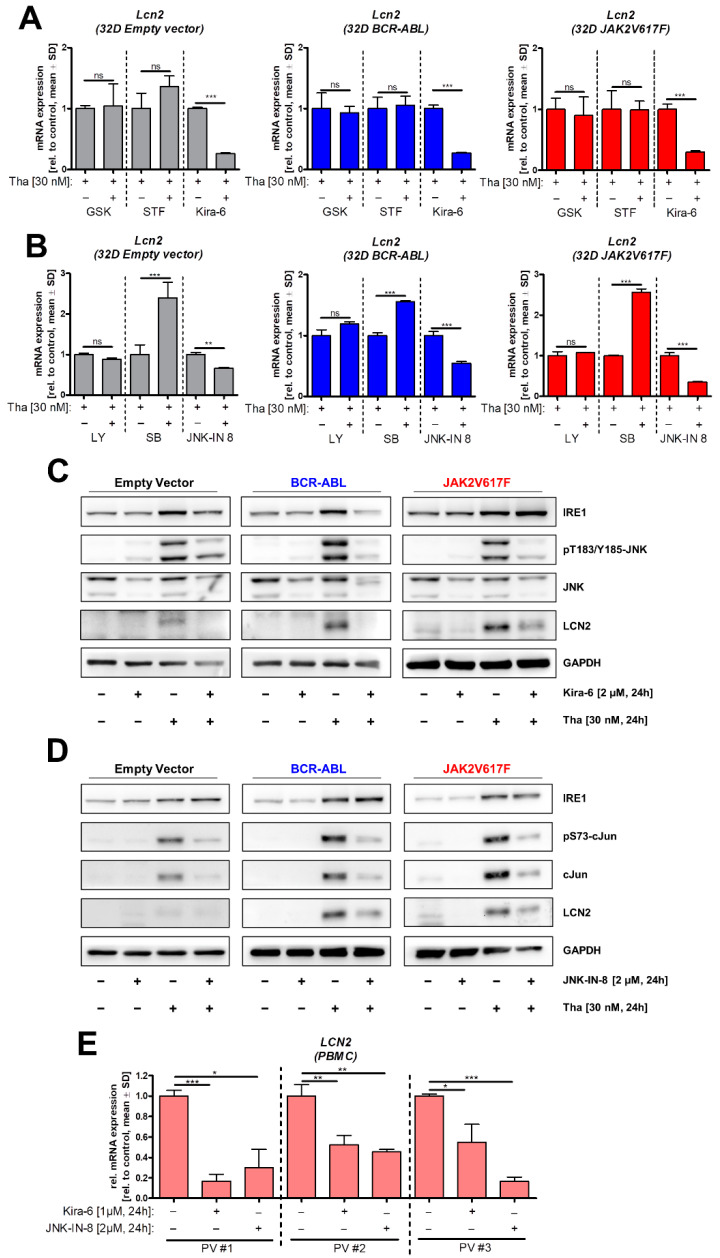
The IRE1–JNK axis regulates Lcn2 induction upon UPR activation. (**A**,**B**) RT-qPCR analysis of 32D cells treated for 24 h with 30 nM Tha in combination with 100 nM GSK2606414 (GSK), 15 µM STF-083010 (STF), 2 µM Kira-6, 5 µM LY2409881 (LY), 5 µM SB203580 (SB) or 2 µM JNK-IN-8. *n* = 3, *t*-test was used to assess differences. (**C**,**D**) Western blot analysis of 32D cell lysates after treatment with 30 nM Tha in combination with 2 µM Kira-6 or 2 µM JNK-IN-8 for 24 h. *n* = 3 (**E**) RT-qPCR analysis of RNA isolated from PBMCs of patients diagnosed with polycythemia vera (PV). Cells were treated for 24 h with 1 µM Kira-6 or 2 µM JNK-IN-8 and mRNA was isolated. *n* = 3, Kruskal-Wallis followed by Dunn´s test. Asterisks indicate *p*-values of: * = *p* < 0.05; ** = *p* < 0.01; *** = *p* < 0.001. ns—not significant.

**Figure 5 cancers-13-04210-f005:**
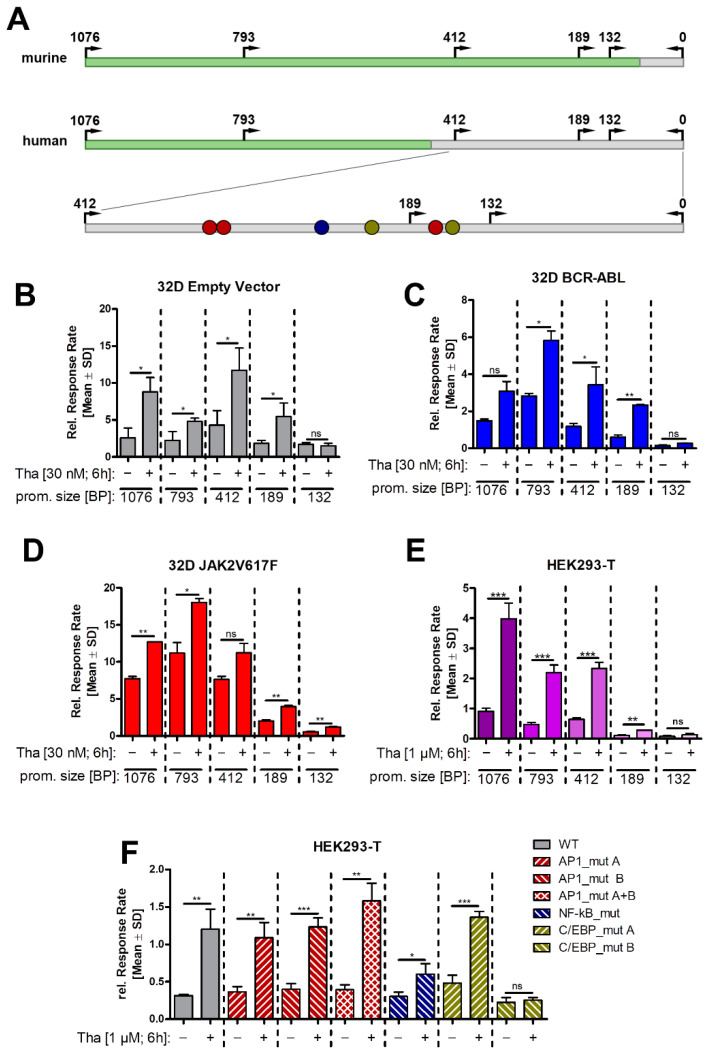
*LCN2* promoter activity is induced by UPR activation. (**A**) Schematic representation of the murine and human *LCN2* promoter region. Arrows indicate fragment size used in the luciferase assays. green = untranscribed region, grey = untranslated region, red dot = AP1 binding site, blue dot = NF-κB binding site, yellow dot = C/EBP binding site. (**B**–**D**) Luciferase assay of 32D cells transfected with pGL3 constructs containing fragments of the murine *Lcn2* promoter. Fragment length in base pairs (bp) as depicted in (**A**). Cells were treated with 30 nM Tha for 6 h prior to analysis. *n* = 3, *t*-test. (**E**) Luciferase assay of HEK293-T cells using pGL3 constructs containing human *LCN2* promoter fragments as depicted in (**A**). Cells were treated with 1 µM Tha for 6 h prior to analysis. *n* = 3, *t*-test. (**F**) Luciferase assay of HEK293-T cells transfected with different mutants of the 412 bp pGL3 construct. Cells were treated with 1 µM Tha for 6 h prior to analysis. *n* = 3, *t*-test. Asterisks indicate *p*-values of: * = *p* < 0.05; ** = *p* < 0.01; *** = *p* < 0.001. ns—not significant.

## Data Availability

All data analyzed during this study are included in this published article (and its Appendix A). Specific requests can be made via E-mail to the corresponding author.

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
