# Peer review of "The Unfolded Protein Response Is a Major Driver of LCN2 Expression in BCR–ABL- and JAK2V617F-Positive MPN"

_cancers, 2021, doi:10.3390/cancers13164210_

Round 1
Reviewer 1 Report
In this work, Tillmann et al studied the mechanisms driving LCN2 over-expression in MPN. This study identified the UPR as the main pathway increasing LCN2 expression, with a contribution of the axis IRE1-JNK-NFKB-C/EBP. Although a lot of work has been performed, I have several concerns regarding the findings of this study.
1- To demonstrate the clinical relevance of studying LNC2 in MPN, the authors tried to reproduce previous results of the literature: LCN2 is overexpressed in MPN patients. The reliability of the cohort cannot be evaluated since the authors do not show the characteristics of the patients. Could the authors give more details about the patients (major parameters of the blood counts, type of MPN, treatment at the time of sampling, driver mutation). Given LCN2 is an inflammatory factor, the inflammatory state of patients should be added to patients characteristics (C-Reactive Protein for example).
Moreover, 2 points made their observations hard to interpret:
- most of the patients were treated when cells and serum were sampled. The authors acknowledged this point, showing that these treatment decreased LCN2 levels in the serum. However, it could be of great interest to clearly show the results of untreated patients.
- mononuclear cells were purified before analysis. Once again, the authors pointed to this weakness and tried to overcome this point by evaluating the expression of MPL, MPO and EPOR to “correct” their results and estimate the proportion of the different lineages present in the MNC population. I don’t think that this analysis is relevant. For example, does MPL expression represent the megakaryocytic compartment or the proportion of hematopoietic stem and progenitor cells that also express this receptor? The proportion of blasts should be included in the characteristics of the patients. How could the author integrate the level of expression of these markers that could be different in the same cell lineages between patients (especially for treated patients)? Another question, how the authors could be sure that the mutation is indeed present in the cells that they studied? Were the JAK2 and CALR allelic burdens determined on the same type of material? If the authors want to demonstrate that MPN cells indeed overexpress LNC2, they should study a cell population that carry the driver mutation such as PMN or CD34+ cells differentiated into granulocytic cells. In addition, the authors performed regressions to correlate LNC2 expression levels to the expression of MPL, MPO and EPOR. How did these regressions were performed? This point is not detailed in the material and method part of the manuscript. More over, how do the author interpret these data? Do the absence of correlation between MPO and LNC2 expression should be interpret as an absence of overexpression of LNC2 in myeloid cells in MPN?
Because the authors failed to clearly demonstrate that hematopoietic cells in MPN overexpressed LCN2, another explanation could be that LCN2 is overexpressed by the bone marrow microenvironment.
2- The authors show a correlation between JAK2/CALR allelic burdens and LCN2 levels in the serum of patients. They also show a correlation between neutrophils counts and LCN2 expression. Are these 2 correlations independent? Could the correlation between allelic burden and LCN2 expression be due to a correlation between neutrophil counts and the allelic burdens? If so, increased levels LCN2 would only be a reflect of the leukocytosis.
3- Why did the authors did not study serum samples of CML patients? Given the results of Figure 1 and 3, it would have been of great interest.
4- To study more precisely the mechanisms that control LNC2 expression in MPN, the authors generated a cell line model based on 32D. I believe that using only one cell line that is a murine one is not adapted. Could the authors confirm their results on 1 or 2 hematopoietic cell lines of human origin? K562 of LAMA84 would be more appropriate for BCR-ABL. HEL, TF1 or UT7 could be interesting for JAK2V617F.
5- Figure 3B suggests that LCN2 is secreted massively from 32D cells. Could BCR-ABL and JAK2V617F have an effect on LCN2 secretion that could explain the increase of LCN2 protein levels in patients’ serum?
6- In figure 3C, the authors exclude that cytokines could increase LCN2 expression. Are they sure that the cytokines were effective on their cell line? It not detailed in the material and method if the cytokines used are murine or human ones. Are their receptors expressed on 32D cells? Are the downstream pathways correctly activated in the cell line after treatment?
7- How the authors explain that IFNa increases LCN2 expression in 32D-JAK2V617F cells (even if not statistically significant), while it decreases LCN2 levels in patients? This raises the reliability of the cell line model used in the study.
8- In figure 3D, the 32D-BCR-ABL cells seem not to express LCN2 anymore in the absence of Thapsigargin (compared to Fig 3A). How to explain this point?
9- The authors observed that a treatment by thapsigargin increased LCN2 expression. Even if thapsigargin indeed triggered an ER stress, how could the authors be sure that LCN2 expression induced by thapsigargin is not due to alteration of calcium signaling? A treatment by other drugs known as ER stress inducers (such as Tunicamycin) should be performed.
10- The authors say that thapsigargin did not induce apoptosis at the doses used. This is not possible to state this because the authors did not used annexin V labeling. PI only labels dead cells. The absence of apoptosis should be studied by combining annexin V labeling to PI (or DAPI) in order to detect cells that are undergoing apoptosis, but are not already dead cells.
Two additional comments:
- Could the authors indicate the number of replicates in each experiment?
- Could the authors indicate the statistical test used in the figure legends
Author Response
Answers to Reviewer 1
We thank the reviewer for his thorough review of our data. As stated below, we have now addressed the mentioned questions and hope to have eliminated any concerns on our study.
1- To demonstrate the clinical relevance of studying LNC2 in MPN, the authors tried to reproduce previous results of the literature: LCN2 is overexpressed in MPN patients. The reliability of the cohort cannot be evaluated since the authors do not show the characteristics of the patients. Could the authors give more details about the patients (major parameters of the blood counts, type of MPN, treatment at the time of sampling, driver mutation). Given LCN2 is an inflammatory factor, the inflammatory state of patients should be added to patients characteristics (C-Reactive Protein for example).
We thank the reviewer for this comment. We have now added a table of the available patient data for the PBMCs and BMMCs including type of MPN, JAK2V617F allele burden, and blood counts (new Table S1). For BMMC sampling, all patients were untreated at that time. In addition, we have added another table for the patient serum samples used in Figure 2 (new Table S2).
Moreover, 2 points made their observations hard to interpret:
- most of the patients were treated when cells and serum were sampled. The authors acknowledged this point, showing that these treatment decreased LCN2 levels in the serum. However, it could be of great interest to clearly show the results of untreated patients.
As mentioned in our answer to question 1, we have added two more tables with patients’ data (new Tables S1 and S2). For BMMCs, all patients were untreated at the time of sampling. Hence, the demonstrated data in Figure 1 are irrespective of any treatment. All used CML samples are from the time of first diagnosis.
For serum, the sample size of untreated patients (“watchful waiting” management) is n = 17. Below, you will find an analysis of serum LCN2 in the untreated Ph- MPN patients in comparison to healthy donors (HD). Although the sample size in each MPN entity is smaller now (except for MF), PV and MF still showed significantly upregulated LCN2, while in ET, LCN2 serum levels were similar to HD. We integrated this figure into the supplementary data (Figure S1B) and describe the novel findings in lanes 282-229. Together, with the data in Figure S1A, demonstrating that the treatment with TKI, IFNa, and HU significantly reduced the amount of LCN2 in the serum, we provide evidence for upregulation of LCN2 protein in MPN and the normalization upon treatment.
Figure for Review 1: New Figure S1B comparing the LCN2 serum level of healthy donors (HD) with untreated ( WW - watch and wait) MPN patient samples divided by their MPN entity.
- mononuclear cells were purified before analysis. Once again, the authors pointed to this weakness and tried to overcome this point by evaluating the expression of MPL, MPO and EPOR to “correct” their results and estimate the proportion of the different lineages present in the MNC population. I don’t think that this analysis is relevant. For example, does MPL expression represent the megakaryocytic compartment or the proportion of hematopoietic stem and progenitor cells that also express this receptor?
Indeed, we tried to correlate the amount of measured LCN2 with gene expression of markers semi-specific for the targeted cell populations, as MNCs were analyzed. We agree that this is only an approximation to the exact composition of the bone marrow samples. However, and although MPL is expressed on HSCs and megakaryocytes (MEPs), we were able to narrow down the LCN2 expression profile to cells not expressing MPL and EPOR. We are convinced that these are valuable data and have added further discussion on this topic into our manuscript (lanes 315-321).
The proportion of blasts should be included in the characteristics of the patients.
We have now included the available data on blasts in patients at CML diagnosis into Table S1.
How could the author integrate the level of expression of these markers that could be different in the same cell lineages between patients (especially for treated patients)?
As discussed above, we generated these data to have an approximation of the cellular composition in relation to LCN2 expression. Importantly, all patients and the respective samples included in this correlation in Figure 1 are untreated. Hence, despite some limitations, the data reflect LCN2 expression in this patient population, but we have included the thoughts and limitations in the discussion of the manuscript (lanes 398-406).
Another question, how the authors could be sure that the mutation is indeed present in the cells that they studied? Were the JAK2 and CALR allelic burdens determined on the same type of material? If the authors want to demonstrate that MPN cells indeed overexpress LNC2, they should study a cell population that carry the driver mutation such as PMN or CD34+ cells differentiated into granulocytic cells.
Indeed, we cannot state that we analyzed solely cells carrying the respective driver mutation. However, this was also not our aim, since the message of differentially increased LCN2 in the MPN entities is not influenced hereby, and differentiation of CD34+ mutated cells into granulocytic cells was beyond the scope of this manuscript. In addition to the primary cell data, we used the 32D cell model to study a cell population which is 100%-positive for the analyzed oncogene.
We have included the allele burden of the BMMCs of Ph- MPN patients analyzed in Figure 1 into Table S1. The allele burden of all samples analyzed in Figure 2 (PV, ET, MF; Table S2) is correlated with the serum levels of LCN2 and is shown in original Figure 2C.
In addition, the authors performed regressions to correlate LNC2 expression levels to the expression of MPL, MPO and EPOR. How did these regressions were performed? This point is not detailed in the material and method part of the manuscript.
Graphpad Prism software was used to perform the regression analyses. We have now included more details in the methods part (lane 186-187), and we apologize for this shortcoming.
Moreover, how do the author interpret these data? Do the absence of correlation between MPO and LNC2 expression should be interpret as an absence of overexpression of LNC2 in myeloid cells in MPN?
We thank the reviewer for raising these important questions, which we have now addressed in the discussion (lanes 398-406). We analyzed mononuclear cells, and terminally differentiated granulocytic cells (neutrophils), the main source of LCN2, are excluded, which may explain the missing correlation of MPO and LCN2. Furthermore, the fluctuating cell composition in the disease entities may hamper the analysis, and the conclusion that LCN2 is not overexpressed in the myeloid compartment cannot be made.
Because the authors failed to clearly demonstrate that hematopoietic cells in MPN overexpressed LCN2, another explanation could be that LCN2 is overexpressed by the bonemarrow microenvironment.
Lu and colleagues demonstrated that LCN2 enhances bone marrow stroma cell proliferation and cell fate determination but expression of LCN2 in the non-hematopoietic compartment was not analyzed (Lu M et al, Blood, 2015; main manuscript citation No.14). The involvement of the bone marrow microenvironment and the expression of LCN2 by MSCs, endothelial cells etc. in MPN background would be interesting to analyze but is beyond the scope of our present work. In lanes 508-512, we already described this issue and added further discussion in lanes 408-411.
We would like to draw the attention to Figure 1A, demonstrating that mononuclear hematopoietic cells in CML, PV and MF show significantly increased LCN2 expression in comparison to healthy control samples.
2- The authors show a correlation between JAK2/CALR allelic burdens and LCN2 levels in the serum of patients. They also show a correlation between neutrophils counts and LCN2 expression. Are these 2 correlations independent? Could the correlation between allelic burden and LCN2 expression be due to a correlation between neutrophil counts and the allelic burdens? If so, increased levels LCN2 would only be a reflect of the leukocytosis.
As suggested by the reviewer, we have now performed correlations of allele burden vs. neutrophils and allele burden vs. leukocytes, and both showed a strong positive correlation. We have included these correlations into new Figure S1E and S1F and described the correlations in the result part of the manuscript (lanes 229-231). High levels of LCN2 correlates with a high amount of leukocytes/neutrophils (Figure 2C and Figure S1G), and leukocytosis correlates with a higher allele burden (new Figure S1E). So, increased levels of LCN2 could reflect leukocytosis. We have added this hypothesis to the discussion section of the manuscript (lanes 415-418).
3- Why did the authors did not study serum samples of CML patients? Given the results of Figure 1 and 3, it would have been of great interest.
We agree with the reviewer that the analysis of CML patients’ serum would be of interest. Our manuscript comprises a high amount of patient samples for the analysis of LCN2 expression and secretion, but another focus is the regulation of LCN2 expression by ER stress via the IRE1–JNK–NF-κB–C/EBP axis. The data of the immunoplex assay in Figure 2 are components of a larger study of secreted proteins in Philadelphia chromosome-negative MPN. We do not have the same data for CML patients and would not be able to collect enough samples of CML at first diagnosis within the deadline provided by the editor. Therefore, we hope that the reviewer agrees with leaving the current data on LCN2 secretion as it is.
4- To study more precisely the mechanisms that control LNC2 expression in MPN, the authors generated a cell line model based on 32D. I believe that using only one cellline that is a murine one is not adapted. Could the authors confirm their results on 1 or 2 hematopoietic celllines of human origin? K562 of LAMA84 would be more appropriate for BCR-ABL. HEL, TF1 or UT7 could be interesting for JAK2V617F.
We performed several assays with HEL and K562 cells, but the data were too variable. This may have been due to variable LCN2 expression in bulk cultures, as observed in a study by Leng and colleagues (Leng X et al, Oncogene, 2008), who demonstrated there was high variability of LCN2 expression in single clones after single cell dilutions of K562 cells. Our 32D cell lines expressing the JAK2V617F or BCR-ABL oncogenes (or empty vector) are clonal and provided reliable data. We therefore used these cells. In addition, for human cells, we used primary cells (PBMCs), readily showing LCN2 expression as well as its regulation by IRE1 and JNK (Figure 4E, showing PBMCs of three PV patients and LCN2 expression regulated by IRE1 and JNK using the inhibitors Kira-6 or JNK-IN-8).
5- Figure 3B suggests that LCN2 is secreted massively from 32D cells. Could BCR-ABL and JAK2V617F have an effect on LCN2 secretion that could explain the increase of LCN2 protein levels in patients’ serum?
The accelerated secretion of LCN2 in BCR-ABL and JAK2V617F-expressing cells is possible, explaining the increased levels of LCN2 in serum of MPN patients. But it could also be a direct association with increased expression. We thank the reviewer for this comment and included a sentence into the discussion of the manuscript highlighting these possibilities (lanes 423-427).
6- In figure 3C, the authors exclude that cytokines could increase LCN2 expression. Are they sure that the cytokines were effective on their cellline? It not detailed in the material and method if the cytokines used are murine or human ones. Are their receptors expressed on 32D cells? Are the downstream pathways correctly activated in the cellline after treatment?
Indeed, the information on the source of the used cytokines was missing. Therefore, we have now added these data to the Materials and Methods section (lanes 118-120). We used recombinant murine TNFa and IFNa and rat IFNg. IL-1b was of recombinant human origin showing cross-reactivity. We regularly use recombinant murine IFNa in our 32D cell models, as described in Czech et al [1] and Schubert et al [2]. We tested the activity of TNFa and IL1ß in our 32D cell model and analyzed activation of NFkB (see below for 32D JAK2V617F cells). TNFa (20 ng/ml) as well as IL-1b (20 and 50 ng/ml) treatment for 30 min effectively phosphorylated p65.
In fact, we observed an increase of LCN2 expression after IFNa and TNFa treatment (Figure 3C; not significant), but ER stress induced by thapsigargin was way more potent in this regard.
Figure for Review 2: 32D JAK2V617F cells (5x105/ml) were treated with the indicated concentrations of TNFa or IL-1b for 30 min. Afterwards, lysates were prepared, and SDS-PAGE and Western Blotting was performed. PVDF membrane was stained with antibodies detecting pSer536-p65 and p65. GADPH served as loading control.
7- How the authors explain that IFNa increases LCN2 expression in 32D-JAK2V617F cells (even if not statistically significant), while it decreases LCN2 levels in patients? This raises the reliability of the cellline model used in the study.
This is certainly an important question. Interferons are inflammatory cytokines and are described to induce LCN2 expression (mainly described for IFNg [3]). We only used short-term treatment of our 32D cell line model, which mimics the inflammatory activity of IFNa, leading to the formation of phosphorylated and transcriptionally active STAT1 homodimers. In MPN patients, IFNa mainly acts on the malignant clone, inducing entry into the cycle, making the malignant clone targetable by the immune system as well as other drugs (Essers MA, Nature 2009 [4] on normal HSCs; Jayaranjan et al. [5]; EHA abstract 2021). The slow reduction of the MPN clone and concomitantly the mutated allele burden may finally lead to a reduction of the LCN2 levels.
8- In figure 3D, the 32D-BCR-ABL cells seem not to express LCN2 anymore in the absence of Thapsigargin (compared to Fig 3A). How to explain this point?
Expression of LCN2 in BCR-ABL and JAK2V617F-expressing 32D cells is still higher in comparison to EV cells. Thapsigargin was a strong inducer of LCN2 expression, so we needed to split the y axis to make the differences visible (new Figure 3D).
9- The authors observed that a treatment by thapsigargin increased LCN2 expression. Even if thapsigargin indeed triggered an ER stress, how could the authors be sure that LCN2 expression induced by thapsigargin is not due to alteration of calcium signaling? A treatment by other drugs known as ER stress inducers (such as Tunicamycin) should be performed.
We performed tunicamycin treatments (see below) and observed less induction of LCN2 expression in comparison to thapsigargin and brefeldin A. We excessively analyzed the signaling pathways activated upon ER stress and confirmed the dependence of LCN2 expression on the IRE1–JNK–NF-κB–C/EBP axis. As we cannot completely exclude the effect of alterations in calcium signaling due to thapsigargin treatment, we have now included this point in our discussion (lanes 485-487).
Figure for Review 3: 32D empty vector, BCR-ABL and JAK2V617F cells were treated with the indicated concentrations of brefeldin A, tunicamycin or thapsigargin for 16h. Subsequently, cells were harvested and RNA was isolated for analysis of Lcn2 expression. Relative mRNA expression of Lcn2 to Gapdh is given.
10- The authors say that thapsigargin did not induce apoptosis at the doses used. This is not possible to state this because the authors did not used annexin V labeling. PI only labels dead cells. The absence of apoptosis should be studied by combining annexin V labeling to PI (or DAPI) in order to detect cells that are undergoing apoptosis, but are not already dead cells.
We agree with the reviewer and have now corrected the sentence in lanes 276-278 to “It should be noted that neither of these concentrations led to an increase of the dead cell fraction (PI-positive) in the analyzed cells after 24 h (Figure S2). After 48 h, higher concentrations of Tha increased the PI-positive cell fraction.” In addition, we adjusted the M&M section in lanes 143-145.
Two additional comments:
- Could the authors indicate the number of replicates in each experiment?
We have now added this information to all figure legends. As stated in the Materials and Methods section, all experiments were performed independently at least three times (lane 189-190).
- Could the authors indicate the statistical test used in the figure legends
As suggested by the reviewer, we have now added the applied statistical tests to the figure legends.

Reviewer 2 Report
This is a well written manuscript that underscores a potential new pathway in MPNs which can be therapeutically targeted. In this manuscript the authors first confirm overexpression of LCN-2 in all myeloproliferative neoplasms except ET through the use of patient samples. They then uses cell line (32D) transfected with either bcr-abl or JAK2 to decipher the regulatory mechanisms behind LCN-2 expression. They show a link between LCN-2 and granulocyte expression and go on to show LCN-2 expression is linked to endoplasmic reticulum stress and the unfolded protein response.
The authors provide sufficient evidence to further explore the IRE1-JNK-NF-kB-C/EBP axis and LCN-2 as a target for novel therapeutics in MPN, especially in MF.
Minor comments:
1. Why were CML patient samples used for studies in Figure 1 but not Figure 2? It would be interesting in Figure 2 to depict normals versus PV, ET, MF and CML and then to evaluate normals vs CALR, JAK2, and bcr-abl. Particularly given the role bcr-abl plays in later studies.
2. Similarly why was CALR not used for the experiments in Figure 3? They mention that only JAK2 and bcr-abl were transfected into 32D cells but a statement why CALR (or MPL for that matter) was not would suffice.
3. While inhibition of the IRE1-JNK-NF-kB-C/EBP axis and its effects on LCN2 would be useful but is not necessary in my opinion for this manuscript. If these experiments were performed then I would include them.
Author Response
We highly appreciate the opinion of the reviewer and are happy to answer the questions below.
Minor comments:
Why were CML patient samples used for studies in Figure 1 but not Figure 2? It would be interesting in Figure 2 to depict normals versus PV, ET, MF and CML and then to evaluate normalsvs CALR, JAK2, and bcr-abl. Particularly given the role bcr-abl plays in later studies.
The immunoplex data on the LCN2 serum levels was acquired in a large screen of the secretome of Philadelphia chromosome-negative MPN. Therefore, we do not have the analysis of CML serum. We agree with the reviewer (and reviewer 1) that the LCN2 levels in the serum of CML patients would be of great interest, but this analysis was beyond the scope of our manuscript and should be analyzed in upcoming studies.
Similarly why was CALR not used for the experiments in Figure 3? They mention that only JAK2 and bcr-abl were transfected into 32D cells but a statement why CALR (or MPL for that matter) was not would suffice.
We thank the reviewer for this question. We decided to concentrate on the most frequent mutation in Ph- MPN, the JAK2V617F mutation. And even though CALR mutated serum samples showed an increase of LCN2 (not significant) in comparison to HD, the increase in JAK2V617F-positive MPN was stronger (here significant) (Figure 2B).
However, we would hypothesize that the mechanism of increased LCN2 may be similar in MPL- or CALR-mutated MPN. As suggested by the reviewer, we have now added this hypothesis to the manuscript (lanes 425-427).
While inhibition of the IRE1-JNK-NF-kB-C/EBP axis and its effects on LCN2 would be useful but is not necessary in my opinion for this manuscript. If these experiments were performed then I would include them.
Until now, we have analyzed the changes of LCN2 expression in three PV patient PBMC samples upon IRE1 (Kira-6) and JNK (JNK-IN-8) inhibition (Figure 4 E). In follow-up studies, targeting one or the other protein in the IRE1-JNK-NF-kB-C/EBP axis in MPN and subsequent analysis of apoptosis induction as well as changes in mutant allele burden in relation to LCN2 expression will be of great interest, which could be followed by in vivo analyses in order to evaluate potential novel therapeutic options for MPN patients.
References cited in this response letter:
1. Czech, J.; Cordua, S.; Weinbergerova, B.; Baumeister, J.; Crepcia, A.;
Han, L.; Maié, T.; Costa, I.G.; Denecke, B.; Maurer, A.; et al. JAK2V617F
but not CALR mutations confer increased molecular responses to
interferon-α via JAK1/STAT1 activation. Leukemia 2019, 33, 995–1010.
2. Schubert, C.; Allhoff, M.; Tillmann, S.; Maié, T.; Costa, I.G.; Lipka, D.B.;
Schemionek, M.; Feldberg, K.; Baumeister, J.; Brümmendorf, T.H.; et al.
Differential roles of STAT1 and STAT2 in the sensitivity of JAK2V617F-
vs. BCR-ABL-positive cells to interferon alpha. J. Hematol. Oncol. 2019,
12, 36.
3. Zhao, P.; Elks, C.M.; Stephens, J.M. The induction of lipocalin-2 protein
expression in vivo and in vitro. J. Biol. Chem. 2014, 289, 5960–9.
4. Essers, M.A.G.; Offner, S.; Blanco-Bose, W.E.; Waibler, Z.; Kalinke, U.;
Duchosal, M.A.; Trumpp, A. IFNalpha activates dormant haematopoietic
stem cells in vivo. Nature 2009, 458, 904–8.
5. Jayarajan, J.; Knoch, J.; Ball, M.; Michael Milsom Interferon-alpha
treatment results in the depletion of dormant JAK2-mutant HSC in a
murine model of polycythemia vera. EHA Congr. 2021 2001, S195.
Round 2
Reviewer 1 Report
The authors have responded to most of my questions.
I still have 3 requests/remarks :
- I would appreciate the authors to add their "figure for review 2" in the supplemental material. If necessary the number of experiments using tunicamycin must be improved to obtain significant results with brefeldin and tunicamycin.
- Statistical tests used are not always appropriate. In particular, in figures 3, ANOVA cannot be performed because gaussian distribution is not assumed with such a low number of replicates. For the same reason, in figure 4 and 5, t-test cannot be performed and should be replaced by Mann&Whitney U-test (as in figure 1).
- In tables S1 and S2, none of the PV patients do not present any obvious polycytemia. Were the patients treated by phlebotomy ? The author must remove the CML patient that present 91% of blasts in the BM (this patients present an acute leukemia even if secondary to CML).
Some other minor points :
- In figure 3D, the axis has not been splitted.
- Figure 4D : I think that there is a mistake in the legend. Cells must have been treated with JNK-IN8
Round 3
Reviewer 1 Report
The authors responded to all my questions.
Author Response
We are glad to have answered all questions satisfactorily. The manuscript was improved by adressing the reviewers comments.